# CARE-RFT: Confidence-Anchored Reinforcement Finetuning for Reliable Reasoning in Large Language Models

## Abstract

Reinforcement finetuning (RFT) has emerged as a powerful paradigm for unlocking reasoning capabilities in large language models. However, we identify a critical trade-off: while unconstrained RFT achieves strong reasoning performance, it severely compromises model trustworthiness by amplifying hallucination and worsening calibration; conversely, RKL-constrained RFT preserves trustworthiness but limits reasoning gains due to its unbounded penalty on exploratory deviations. To resolve this tension, we introduce **CARE-RFT** (Confidence-Anchored Regularized Reinforcement Finetuning), a novel method that replaces standard reverse KL regularization with a skew reverse KL divergence. CARE-RFT provides a *confidence-sensitive* penalty—bounded for confident, consistently-rewarded explorations to enable reasoning, while unbounded elsewhere to preserve calibration. Extensive experiments across multiple model scales and RFT algorithms show that CARE-RFT achieves a superior balance, matching the reasoning performance of unconstrained RFT while recovering the trustworthiness and calibration of the base model. Our work establishes that careful, confidence-aware regularization is key to building both capable and trustworthy reasoning models.

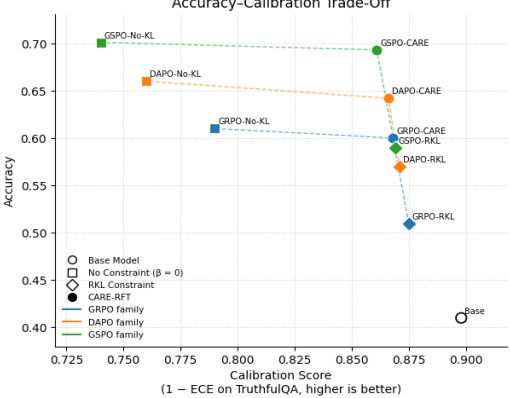

Figure 1: **CARE-RFT breaks the accuracy–calibration trade-off.** Across GRPO, DAPO, and GSPO on Qwen2.5-3B, unconstrained RL boosts accuracy but destroys calibration on MATH (Hendrycks et al., 2021) and TRUTHFULQA (Lin et al., 2021), while RKL restores calibration at the cost of accuracy. **CARE-RFT consistently moves each method toward the upper-right region**—achieving strong reasoning gains *and* stable factual reliability.

## 1 Introduction

Large reasoning models have advanced rapidly, with a landmark example being DeepSeek-R1-Zero (Guo et al., 2025), which established reinforcement finetuning (RFT) as a powerful post-training paradigm. By applying RL directly to base models, RFT helps elicit language models' emergent behaviors such as step-by-step analysis, self-reflection, and backtracking (the "aha moment") (Guo et al., 2025; Huan et al., 2025). At its core is Group Relative Policy Optimization (GRPO) (Shao et al., 2024), an algorithm that avoids the computational overhead of learning explicit value and

reward models. Several variants of GRPO have since been proposed to further improve performance and stability (Yu et al., 2025; Liu et al., 2025; Zheng et al., 2025; Zhao et al., 2025).

While these advances have significantly improved model reasoning performance, their impacts on model trustworthiness remain underexplored (Huang et al., 2025; 2024; Yao et al., 2025). A central concern is hallucination (Song et al., 2025; Farquhar et al., 2024; Kalai et al., 2025): RFT-trained models often perform poorly on fact retrieval and may produce fabricated answers to ambiguous, under-specified (Yin et al., 2023), or unanswerable questions (Sun et al.; Song et al., 2025). Such behavior poses serious risks in high-stakes domains such as healthcare, law and finance (Asgari et al., 2025; Kim et al., 2025). Evidence from both industry and academia further shows that more capable reasoning models can hallucinate more severely (OpenAI, 2024; Hughes et al., 2023; Yao et al., 2025; Song et al., 2025; Kalai et al., 2025), challenging the assumption that stronger reasoning performance translates into greater model reliability (Huan et al., 2025). Despite this, there remains a lack of focused investigation into hallucination under RFT.

In this paper, we investigate the hallucination pitfalls of reinforcement finetuning and propose a simple, principled remedy. Our first empirical observation is that recent RFT variants (Yu et al., 2025; Zheng et al., 2025; Liu et al., 2025; Zhao et al., 2025) that *omit* reverse-KL (RKL) (Kullback, 1951) regularization achieve stronger reasoning accuracy yet exhibit higher hallucination rates and larger Expected Calibration Error (ECE) on fact-seeking benchmarks than their KL-regularized counterparts. This happens because only outcome-level supervision—propagating a single response-level signal *uniformly* to all tokens—is available in constraint-free RFT. This coarse credit assignment amplifies hallucination and worsens calibration (Section 3.1).

To obtain a better-calibrated model, the trained policy should remain close to the base model, which recent studies suggest is relatively well-calibrated (Kalai et al., 2025). A natural approach is to impose a constraint, typically via RKL between $\pi_\theta$ and $\pi_{\text{ref}}$, where $\pi_\theta$ is the policy being optimized and $\pi_{\text{ref}}$ is a fixed reference policy. RKL provides useful *token-level* guidance constraining $\pi_\theta$ to remain close to $\pi_{\text{ref}}$ instead of just applying the same outcome-based reward to all tokens. However, RKL imposes an unbounded penalty in regions where $\pi_{\text{ref}}$ assigns low probability, making it difficult for $\pi_\theta$ to explore novel, high-reward generations that deviate from the reference model. As a result, exploration along promising but low-probability reasoning paths is discouraged, hindering model improvements in reasoning performance (Section 3.2).

These insights motivate **CARE-RFT** (Confidence-Anchored Regularized Reinforcement Finetuning), which replaces standard RKL with a *skew reverse KL* (Lee, 2001) penalty that is confidence-sensitive. It relaxes RKL when $\pi_{\text{ref}}$ has low probability: Skew reverse KL provides bounded penalty on tokens that is consistently rewarded and $\pi_\theta$ is confident about, while enforcing unbounded penalty elsewhere—preserving calibration without sacrificing the exploratory moves needed for reasoning. Empirically, we observe that CARE-RFT achieves a superior trade-off, matching the reasoning performance of unconstrained RFT while recovering the trustworthiness and calibration of the base model (Section 5).

## 2 PRELIMINARIES

We present a unified formulation of reinforcement finetuning (RFT) algorithms that subsumes most variants used in practice. Throughout the paper, we use the following notations: $(q, o)$ denotes a question–response pair drawn from $\mathcal{D}_{\pi_\theta}$. In most settings, $q$ is sampled from a fixed dataset and $o$ is generated by the current policy $\pi_\theta$. The response length is denoted by $|o|$, and $o_t$ and $o_{<t}$ denote the $t$-th token and the tokens preceding the $t$-th token, respectively. The reward $r$ depends on the generated and correct answer.

**Generic RFT objective.** Let $\pi_\theta$ be the behavior policy that generated $o$. We maximize

$$\mathcal{J}_{\mathcal{A}}(\theta) = \mathbb{E}_{(q,o)\sim\mathcal{D}_{\pi_\theta}} \left[ \frac{1}{|o|} \sum_{t=1}^{|o|} \underbrace{C_{\mathcal{A}}\left(q, o, t, r\right)}_{\text{surrogate for algorithm } \mathcal{A}} - \underbrace{\beta \mathcal{D}iv(q, o, t, \pi_{\text{ref}})}_{\text{divergence penalty to } \pi_{\text{ref}}} \right]. \tag{1}$$

The gradient of a generic RFT objective equation 1 with respect to model parameters $\theta$ can be expressed as:

$$\nabla_\theta \mathcal{J}_\mathcal{A}(\theta) = \mathbb{E}_{(q,o) \sim \mathcal{D}_{\pi_\theta}} \left( \frac{1}{|o|} \sum_{t=1}^{|o|} \underbrace{(GC_\mathcal{A}(q,o,t,r) - \beta GC_{\mathcal{D}iv}(q,o,t,\pi_{\text{ref}}))}_{\textit{Gradient Coefficient}} \nabla_\theta \log \pi_\theta(o_t \mid q, o_{<t}) \right).$$

(2)

The *gradient coefficient* decomposes into two terms: (i) $GC_\mathcal{A}$, determined by algorithm $\mathcal{A}$ and the reward signal $r$, and (ii) $GC_{\mathcal{D}iv}$, a divergence-based regularization term that controls $\pi_\theta$'s deviation from a reference model $\pi_{\text{ref}}$. The scaling factor $\beta$ governs the regularizer's strength. In practice, $GC_{\mathcal{D}iv}$ often takes the form of a reverse KL divergence penalty, as in PPO (Schulman et al., 2017) and GRPO (Shao et al., 2024), to mitigate reward hacking and uncontrolled policy drift.

**(Reverse) Kullback–Leibler Divergence.** The divergence between two probability distributions quantifies how dissimilar the distributions are, with KL-based divergences being among the most widely used. The reverse KL (RKL) divergence is

$$D_{\text{RKL}}\big(\pi_{\text{ref}}(o_t \mid q, o_{<t}) \,|\, \pi_\theta(o_t \mid q, o_{<t})\big) = \sum_{o_t \in \mathcal{V}} \pi_\theta(o_t \mid q, o_{<t}) \log \frac{\pi_\theta(o_t \mid q, o_{<t})}{\pi_{\text{ref}}(o_t \mid q, o_{<t})},$$

(3)

where $\mathcal{V}$ is the vocabulary. In practice, since only the sampled token $o_t$ is observed, the training objective uses the per-token estimator $\log \frac{\pi_\theta(o_t \mid q, o_{<t})}{\pi_{\text{ref}}(o_t \mid q, o_{<t})}$.

The gradient of such an objective will be:

$$\nabla_\theta D_{\text{RKL}}(\pi_{\text{ref}}(o_t \mid q, o_{<t}) \,|\, \pi_\theta(o_t \mid q, o_{<t})) = \sum_{o_t \in \mathcal{V}} \underbrace{\left(\log \frac{\pi_\theta(o_t \mid q, o_{<t})}{\pi_{\text{ref}}(o_t \mid q, o_{<t})} + 1\right)}_{\textit{Gradient Coefficient}} \nabla_\theta \pi_\theta(o_t \mid q, o_{<t})$$

(4)

**Expected Calibration Error (ECE).** To quantify model calibration, we adopt the Expected Calibration Error (ECE), a standard metric measuring the discrepancy between predicted confidence and empirical accuracy: for each input $x$, the model generates $N$ responses $\{r_i\}_{i=1}^N$ with extracted answers $\{a_i\}$. The majority-voted answer $a$ has confidence

$$P(a) = \frac{1}{N} \sum_{i=1}^N \mathbb{1}(a_i = a).$$

Given evaluation set $\mathcal{D} = \{(x_j, y_j)\}_{j=1}^n$, correctness is $c_j = \mathbb{1}(a_j \equiv y_j)$. Partitioning $[0,1]$ into $M$ bins $\{B_m\}$, we compute

$$\text{acc}(B_m) = \frac{1}{|B_m|} \sum_{j \in B_m} c_j, \ \text{conf}(B_m) = \frac{1}{|B_m|} \sum_{j \in B_m} P(a_j), \ \text{ECE} = \sum_{m=1}^M \frac{|B_m|}{n} \big|\text{acc}(B_m) - \text{conf}(B_m)\big|,$$

with ECE $= 0$ indicating perfect calibration. We follow Yao et al. (2025) with $N = 10$ samples and $M = 10$ bins.

## 3 FAILURE MODES OF UNCONSTRAINED AND RKL CONSTRAINED RFT

To take a closer look at where and why unconstrained and RKL-constrained RFT fails, we designed a GRPO-based experiment that disentangles the effects of positive and negative rewards on policy learning. Concretely, we construct three variants: *+Reward Update*, which follows the GRPO update but masks out samples with negative advantage values, thereby keeping only terms with positive advantage (i.e., correct samples); *-Reward Update*, which applies the same update but masks out positive-advantage samples instead; and the *Full Update*, which allows both. This separation allows us to isolate the individual impacts of positive versus negative samples on the behavior of the

learned policy. Using these experiments, we show that while unconstrained RFT improves reasoning performance, it also makes the model less calibrated, highlighting the necessity of incorporating constraints during RFT training (Section 3.1). The natural constraint RKL, however, restricts exploration, motivating the need for new forms of constraints that allow for exploration while keeping the model close to the better-calibrated base model (Section 3.2).

## 3.1 Consequences of Unconstrained Optimization

Recent RFT works (Section E) often remove the RKL constraint altogether. The motivation is empirical: during training on long-CoT tasks, strict RKL constraint prevents the model from diverging sufficiently from the base model distribution to unlock emergent reasoning behaviors. However, removing RKL introduces unintended consequences, such as hallucination and poor calibration.

| Method | Qwen2.5-3B-Base | MATH | TruthfulQA | ECE |
|---|---|---|---|---|
| **+Reward Updates** | *No RKL* | 0.50 (↑0.09) | 0.379 (↓0.11) | 0.19 (↑0.088) |
| | *With RKL* | 0.45 (↑0.04) | 0.47 (≃) | 0.142 (↑0.04) |
| **−Reward Updates** | *No RKL* | 0.29 (↓0.12) | 0.31 (↓0.179) | 0.242 (↑0.14) |
| | *With RKL* | 0.40 (↓0.01) | 0.42 (↓0.069) | 0.171 (↑0.069) |
| **Full Updates** | *No RKL* | 0.61 (↑0.2) | 0.35 (↓0.139) | 0.21 (↑0.108) |
| | *With RKL* | 0.51 (↑0.1) | 0.48 (≃) | 0.125 (↑0.023) |

Table 1: Performance of the model trained with positive-only (+Reward), negative-only (−Reward), and full updates in GRPO, with and without RKL regularization, on **MATH** (Hendrycks et al., 2021) (reasoning), **TruthfulQA** (Lin et al., 2021) (fact retrieval), and Expected Calibration Error (ECE; lower is better).

We observe that unconstrained RFT training (*Full Update*) improves reasoning performance (MATH +0.20) but harms factuality (TruthfulQA −0.139) and calibration (ECE +0.108) (Table 1). To understand why unconstrained RFT induces hallucination and miscalibration, we analyze updates restricted to positive samples (+Reward Update) or negative samples (-Reward Update) and uncover two distinct failure modes, which arise from indiscriminate reinforcement and penalization. By "indiscriminate", we refer to the fact that, when the RKL constraint is removed, outcome-based rewards are uniformly applied across all tokens in a generation, thereby reinforcing or penalizing every token indiscriminately rather than selectively adjusting those associated with specific reasoning steps. We next elaborate on these two failure modes.

First, **indiscriminate reinforcement** increases the probability of any generation that happens to receive a high outcome reward, thereby reinforcing the entire chain of thought—even when intermediate steps are logically flawed. Over training, the language model $\pi_\theta$ may concentrate probability mass on a limited set of high-reward generations, which can contain *spurious reasoning steps*—erroneous intermediate steps that nonetheless lead to a correct final outcome. While such steps may improve performance on reasoning tasks, they do not generalize to other settings, resulting in degraded factuality (Table 1: No KL improves MATH by +0.09 but reduces TruthfulQA by −0.11 and increases ECE by +0.088). Consequently, the model becomes increasingly *overconfident*: responses with flawed reasoning are placed into high-confidence bins, leading to miscalibration (Figure 2, +Reward Update).

Second, **indiscriminate penalization** uniformly reduces the probability of every token in generations that yield incorrect answers, regardless of whether individual steps are correct or erroneous. Given a well-trained base model, this will result in primarily down-weighing the probability of correct reasoning steps along with a small number of erroneous ones. Over the training process, the indiscriminate penalization will erode desirable behaviors in the base model such as grammatical correctness, coherence, and factual grounding, leading to a form of *forgetting*. Consequently, the model becomes increasingly *uncertain* and exhibits worse calibration: even correct responses are produced with low confidence (Figure 2: -Reward Update, Table 1: −Reward, No KL lowers TruthfulQA by −0.179, and increases ECE by +0.14). Overall, under unconstrained RFT, indiscriminate penalization incurs greater harm on calibration than indiscriminate reinforcement, as reflected in the higher degradation of TruthfulQA performance under −Reward compared to +Reward.

Taken together, these results show that unconstrained RFT, despite improving a model's reasoning capability, simultaneously undermines its fact retrieval reliability and calibration through two distinct mechanisms. Indiscriminate reinforcement drives the model toward overconfidence in spurious reasoning patterns, whereas indiscriminate penalization induces forgetting. Thus, it is essential to impose learning signals at the token level rather than indiscriminately across all tokens in a generation. This allows different tokens to receive non-uniform reinforcement or penalization. A natural way to provide token-level supervision is through constraints in RFT, since such constraints operate at the token level and assign varying weights across tokens. Moreover, as recent work has shown, base language models are generally well-calibrated (Kalai et al., 2025), and hence constraining the trained policy $\pi_\theta$ to remain close to the reference policy $\pi_{\text{ref}}$ can improve calibration. We next discuss the natural constraint RKL and examine why, although it helps maintain calibration, it may not be ideal for reasoning tasks.

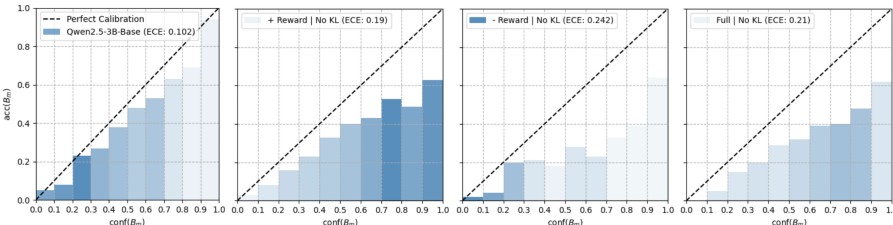

Figure 2: ECE plot comparing base model with its +Reward and -Reward Update checkpoints on TruthfulQA. Each plot visualizes the relationship between model confidence $\text{conf}(B_m)$—estimated via sampling and majority voting—and the actual correctness probability $\text{acc}(B_m)$. Models closer to the diagonal with lower Expected Calibration Error (ECE) are better calibrated. A darker color means more responses are concentrated in this confidence interval.

## 3.2 LIMITATIONS OF REVERSE KL REGULARIZATION

While our analysis indicates that some form of constraint is necessary, a commonly adopted choice is the reverse KL (RKL) divergence (Zheng et al., 2023; Shao et al., 2024). Applied at the token level and providing per-step feedback, RKL mitigates the indiscriminate reinforcement and penalization observed in constraint-free RFT, which depends solely on outcome-based rewards.

A key limitation of RKL is that its penalty is *unbounded*: whenever $\pi_{\text{ref}}(o_t) \to 0$ while $\pi_\theta(o_t)$ remains non-negligible, the gradient coefficient in Eq 4 for the penalty term diverges to $-\infty$. In effect, the constraint overwhelms the reward signal and forbids exploration into any token the reference model assigns low probability. Yet these low-probability regions are precisely where reinforcement finetuning can foster novel reasoning strategies.

Thus, although RKL successfully enables the model to preserve calibration and factuality, its unbounded penalty prevents the policy from accumulating probability mass on novel reasoning generations, leading to limited gains on reasoning benchmarks compared to RL fine-tuning without KL divergence—for example, comparing the "+Reward / No RKL" and "+Reward / with RKL" settings in Table 1 shows that unconstrained RFT yields greater improvements on MATH than RKL-constrained ones.

## 4 CONFIDENCE-ANCHORED REGULARIZED REINFORCEMENT FINETUNING

The analysis above shows that neither the constraint-free nor the RKL constrained RL fine-tuning is satisfactory: the former amplifies spurious trajectories and collapses calibration, while the latter overconstrains probability increases and throttles exploration. The key insight is that reinforcement finetuning demands *direction-sensitive, confidence-anchored regularization*—gentle on probability increases to allow useful reasoning patterns to accumulate, but strict on probability decreases to safeguard pretrained priors. We operationalize this principle through **CARE-RFT** (Confidence-Anchored Regularized Reinforcement Finetuning), which incorporates a skew reverse KL (SRKL) (Lee, 2001) penalty that adapts the divergence term to the model's own confidence. Intuitively, the SRKL term in CARE-RFT anchors the policy closely to the reference in uncertain regions (protecting calibration) while relaxing the anchor when the model is confident and consistently rewarded (enabling reasoning).

The definition of skew reverse KL (Lee, 2001) employs a parameter $\alpha \in (0, 1)$ that controls the mixing ratio of two distributions. The $\alpha$-SRKL between the current policy and the reference policy is defined as the KLD between the current policy and the mixture of distributions. Specifically, the *skew reverse KL* is

$$D_{\mathrm{SRKL}}^{\alpha}\big(\pi_{\mathrm{ref}}(o_t \mid q, o_{<t}) \,\|\, \pi_\theta(o_t \mid q, o_{<t})\big)$$

$$= \sum_{o_t \in \mathcal{V}} \pi_\theta(o_t \mid q, o_{<t}) \log \frac{\pi_\theta(o_t \mid q, o_{<t})}{\alpha\pi_\theta(o_t \mid q, o_{<t}) + (1 - \alpha)\pi_{\mathrm{ref}}(o_t \mid q, o_{<t})}. \tag{5}$$

Its gradient form:

$$\nabla_\theta D_{\mathrm{SRKL}}^{\alpha}\big(\pi_{\mathrm{ref}}(o_t \mid q, o_{<t}) \,\|\, \pi_\theta(o_t \mid q, o_{<t})\big) = \sum_{o_t \in \mathcal{V}} (\log \frac{\pi_\theta(o_t \mid q, o_{<t})}{\alpha\pi_\theta(o_t \mid q, o_{<t}) + (1 - \alpha)\pi_{\mathrm{ref}}(o_t \mid q, o_{<t})}$$

$$+ 1 - \alpha \frac{\pi_\theta(o_t \mid q, o_{<t})}{\alpha\pi_\theta(o_t \mid q, o_{<t}) + (1 - \alpha)\pi_{\mathrm{ref}}(o_t \mid q, o_{<t})})\nabla_\theta \pi_\theta(o_t \mid q, o_{<t}), \tag{6}$$

where the greyed term is the gradient coefficient.

When $\alpha \to 0$, SRKL reduces to standard RKL. As $\alpha$ increases, the effective "anchor" is not fixed but confidence-sensitive: the target distribution becomes a convex combination that includes the current policy, when the current policy is uncertain, the anchor leans toward the reference; when highly confident, the anchor shifts toward the current policy, bounding the penalty on exploratory deviations. Empirically and theoretically, such skewing tames gradient blow-ups and estimator variance compared to standard RKL divergence (details below).

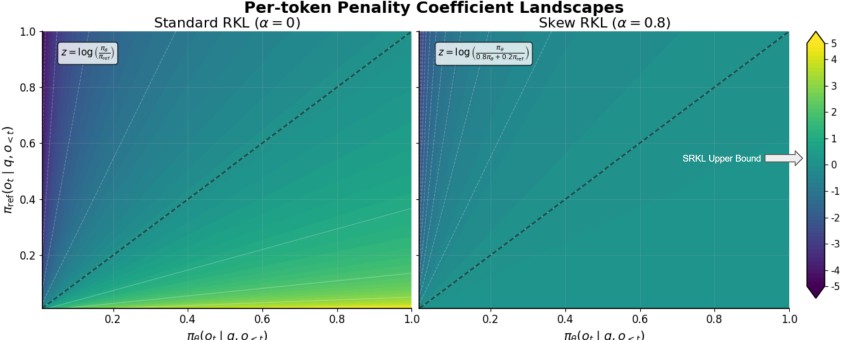

Figure 3: Penalty landscapes for reverse KL ($\alpha = 0$, left) and skew reverse KL ($\alpha = 0.8$, right). Standard RKL imposes unbounded penalties whenever the reference strongly disfavors, while skew RKL introduces a *one-sided bound*: upward deviations are capped by a finite penalty, enabling stable exploration, whereas downward deviations remain strongly penalized to preserve calibration.

The skewed reference $\alpha\pi_\theta + (1 - \alpha)\pi_{\mathrm{ref}}$ induces a *one-sided bounded penalty*: upward moves (increasing probability mass beyond $\pi_{\mathrm{ref}}$) face only a capped cost, allowing consistently rewarded reasoning patterns to accumulate, while downward moves (reducing mass below $\pi_{\mathrm{ref}}$) are effected without bound, preserving pretrained priors.

This one-sided boundedness is evident in the per-token gradient coefficient of Eq. (7).

$$C_\alpha(r) = \log \frac{r}{\alpha r + (1 - \alpha)} + 1 - \alpha \frac{r}{\alpha r + (1 - \alpha)}, \qquad r = \frac{\pi_\theta(o_t \mid q, o_{<t})}{\pi_{\mathrm{ref}}(o_t \mid q, o_{<t})} \geq 0.$$

Unlike reverse KL, where $C_0(r) = \log r + 1$ is unbounded in both directions, we can easily check that $C_\alpha(r)$ admits a finite upper bound. In particular, one can verify that

$$C_\alpha(r) \leq \log \frac{1}{\alpha}$$

with the proof deferred to Appendix A.

This bound guarantees that upward probability shifts cannot be over-penalized, preventing the suppression of novel but correct generations. At the same time, as $r \to 0$ the coefficient diverges negatively, ensuring that downward moves are strongly resisted to safeguard calibration. Figure 3 illustrates this confidence-sensitive asymmetry: unlike RKL, SRKL caps the penalty on exploratory increases while maintaining strong anchoring against unwarranted decreases. This mechanism allows CARE-RFT to retain reasoning gains from exploration without incurring the miscalibration costs of unconstrained updates.

Empirically, the boundedness of $C_\alpha(r)$ manifests as a visible contraction of the gradient-coefficient distribution as $\alpha$ increases (Fig. 4), and corresponds to smoother learning curves and more stable updates. We defer full experimental details and ablations to Sec. 5.

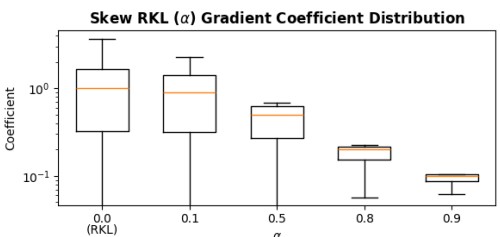

Figure 4: Gradient coefficient distribution for RKL and SRKL across different skew values $\alpha$

In summary, **CARE-RFT** combines outcome-driven learning with a confidence-anchored divergence to preserve calibration while enabling reasoning. Its core regularizer—skew reverse KL (SRKL)—interpolates between the current policy and a fixed reference, which (i) bounds the per-token gradient coefficient (cf. Eq. 6), mitigating low-support explosions that arise with reverse KL; (ii) discourages indiscriminate sharpening in low-confidence regions, thereby preventing hallucination and miscalibration; and (iii) still permits high-confidence, reward-consistent deviations needed for emergent reasoning (see Fig. 3). Practically, this makes CARE-RFT a **drop-in, factuality-enhancing regularization layer** for GRPO-style RFT: it stabilizes optimization, curbs hallucination-prone overconfidence, and maintains the exploratory capacity required to acquire skills beyond the reference model. As shown in §5, these properties hold across GRPO variants and model scales, matching unconstrained RFT on reasoning accuracy while maintaining or even improving calibration and factuality on ambiguity- and retrieval-sensitive benchmarks.

## 5 EXPERIMENT

This section presents a comprehensive evaluation of CARE-RFT. We first benchmark its performance against key RFT algorithms, demonstrating a superior trade-off between reasoning accuracy and trustworthiness. We then analyze training dynamics through token entropy to explain this improvement mechanistically. Finally, we ablate the core hyperparameter $\alpha$ to validate our design choices.

### 5.1 EXPERIMENTAL SETUP

**Models and Baselines.** We conduct experiments on the **Qwen2.5-3B** and **Qwen2.5-7B** base models (Team, 2024). To demonstrate generality, we integrate CARE-RFT into three popular RFT algorithms: **GRPO** (Shao et al., 2024), the foundational method; **DAPO** (Yu et al., 2025), an aggressive constraint-free variant; and **GSPO** (Zheng et al., 2025), a recent sequence-level approach. Using VeRL's (Sheng et al., 2024) implementation of each algorithm, we compare three key configurations: the unconstrained version (**No Constraint**, $\beta = 0$), the standard **RKL Constraint** ($\beta = 0.04$), and our **CARE-RFT** ($\beta = 0.04$, $\alpha = 0.8$). Full training details are provided in the appendix.

**Datasets and Metrics.** We perform reinforcement finetuning on the **MATH** training set (Hendrycks et al., 2021). Performance is then evaluated on the held-out test sets of standard reasoning benchmarks (**MATH** (Hendrycks et al., 2021) and **GSM8K** (Cobbe et al., 2021), reported as accuracy) and hallucination benchmarks (**TruthfulQA** (Lin et al., 2021) and **SelfAware** (Sun et al.), which tests fact retrieval and awareness of unanswerable questions, reported as accuracy). Pass@4 is used for reasoning tasks; Pass@1 for trustworthy tasks. To measure calibration, we report the **Expected Calibration Error (ECE)** (Naeini et al., 2015) on TruthfulQA.

### 5.2 MAIN RESULTS

We first present the overall performance of CARE-RFT compared to constrained and unconstrained RFT baselines. Table 2 reports results on the Qwen2.5-3B model; results on the 7B scale show

consistent trends and are included in the appendix. The key finding is that CARE-RFT consistently achieves a superior balance, matching the reasoning performance of unconstrained RFT while recovering the trustworthiness of base model.

Table 2: **Main Results on Qwen2.5-3B.** Comparison of RFT algorithms with different constraints on reasoning (MATH, GSM8K), factuality (TruthfulQA, SelfAware), and calibration (ECE). CARE-RFT achieves the best trade-off, nearly matching the reasoning scores of unconstrained methods while maintaining strong trustworthiness.

| Method | MATH | GSM8K | SelfAware | TruthfulQA | ECE ↓ |
|---|---|---|---|---|---|
| **Base Model** | 0.410 | 0.791 | 0.372 | 0.489 | 0.102 |
| GRPO (No Constraint) | 0.610 | 0.854 | 0.249 | 0.350 | 0.210 |
| RKL-GRPO | 0.510 | 0.818 | 0.351 | 0.480 | 0.125 |
| **CARE-GRPO** | **0.600** | **0.860** | **0.355** | **0.465** | **0.132** |
| DAPO (No Constraint) | 0.660 | 0.889 | 0.232 | 0.312 | 0.240 |
| RKL-DAPO | 0.570 | 0.8432 | 0.346 | 0.478 | 0.129 |
| **CARE-DAPO** | **0.642** | **0.872** | **0.334** | **0.461** | **0.134** |
| GSPO (No Constraint) | 0.701 | 0.902 | 0.243 | 0.304 | 0.260 |
| RKL-GSPO | 0.590 | 0.8681 | 0.341 | 0.469 | 0.131 |
| **CARE-GSPO** | **0.693** | **0.907** | **0.332** | **0.459** | **0.139** |

### 5.3 ANALYSIS OF TRAINING DYNAMICS

The results in Table 2 establish that CARE-RFT improves the trustworthiness of RFT. We now investigate the mechanistic cause of this improvement by analyzing the evolution of token-level entropy during training. Entropy measures the uncertainty of the model's token-level predictions. A rapid collapse in entropy indicates the model is becoming overconfident, which is a known precursor to miscalibration (Song et al., 2025). **While token entropy is not itself a direct measure of calibration, its collapse is a strong signal of distributional sharpening that typically leads to poor ECE.** We hypothesize that the better ECE of CARE-RFT stems from its ability to prevent such a collapse while still allowing the policy to effectively put high probability mass on correct reasoning paths.

Figure 5 plots the average token entropy for GRPO and its constrained variants on the Qwen2.5-3B model throughout training (analogous plots for DAPO and GSPO are observed). The unconstrained GRPO exhibits a sharp, monotonic decrease in entropy, converging to near-zero values. This indicates a severe collapse of the probability distribution, making the model brittle and overconfident, which directly explains its high ECE (0.210). In contrast, RKL successfully constrains the model, enabling it to maintain at a much higher entropy plateau and thus much better calibration (ECE 0.125). However, this comes at the cost of limiting the model's ability to sharpen its distributions for complex reasoning, resulting in lower MATH accuracy.

Strikingly, CARE-RFT finds an intermediate regime. It permits a significant and useful decrease in entropy compared to the RKL constraint, enabling the strong reasoning performance seen in Table 2. However, it definitively avoids the catastrophic collapse seen in the unconstrained case, stabilizing at a healthy entropy level that is predictive of its low ECE (0.132). This controlled entropy reduction is the core mechanism behind CARE-RFT's ability to balance exploration (needed for reasoning) while constraining its deviance to the base model (needed for calibration).

### 5.4 ABLATION STUDY ON THE SKEW PARAMETER $\alpha$

The previous sections demonstrate that CARE-RFT with $\alpha = 0.8$ achieves an optimal balance. We now ablate the skew parameter $\alpha$ to validate this design choice and understand its sensitivity. Table 3 shows the performance of GRPO on Qwen2.5-3B across different $\alpha$ values, with $\beta$ fixed at 0.04.

The results confirm a clear trend: as $\alpha$ increases from 0 (standard RKL) to 0.8, reasoning performance improves substantially (MATH improves from 0.51 to 0.60) while trustworthiness metrics remain stable. This demonstrates that relaxing the constraint in a confidence-sensitive manner directly enables the reasoning gains we observe. However, when $\alpha$ becomes too large (0.9), the method begins to behave more like the unconstrained case, with reasoning performance plateauing and trustworthiness starting to degrade. This ablation validates that $\alpha = 0.8$ represents a sweet

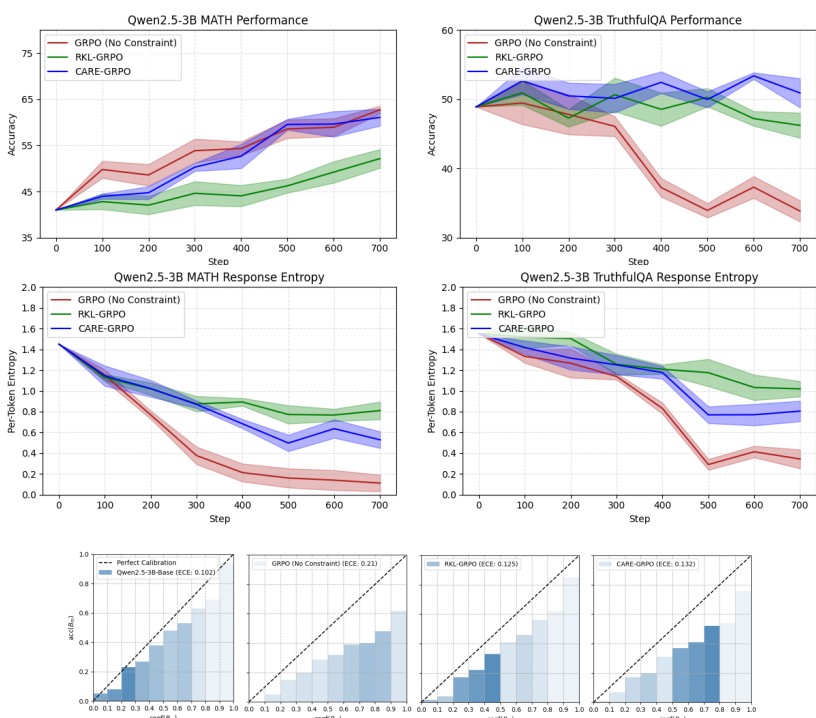

Figure 5: **Token-level entropy during training of GRPO variants on Qwen2.5-3B.** Unconstrained RFT leads to entropy collapse, causing overconfidence and high ECE. RKL prevents collapse but limits performance gains. CARE-RFT allows for controlled entropy reduction, achieving a balance that explains its superior calibration-performance trade-off.

Table 3: **Ablation study of the skew parameter $\alpha$ in CARE-RFT on Qwen2.5-3B.** As $\alpha$ increases from 0 (equivalent to RKL) to 0.8, reasoning performance improves while trustworthiness remains strong. Values beyond 0.8 begin to degrade performance, indicating the method's robustness within a practical range.

| $\alpha$ | MATH | GSM8K | SelfAware | TruthfulQA | ECE $\downarrow$ |
|---|---|---|---|---|---|
| 0.0 (RKL) | 0.510 | 0.818 | 0.351 | 0.480 | 0.125 |
| 0.4 | 0.562 | 0.841 | 0.353 | 0.472 | 0.128 |
| 0.8 | 0.600 | 0.860 | 0.355 | 0.465 | 0.132 |
| 0.9 | 0.592 | 0.855 | 0.349 | 0.458 | 0.138 |

spot in the trade-off, and shows that CARE-RFT is not overly sensitive to precise parameter choices within a reasonable range.

## 6  CONCLUSION

We identified a critical trade-off in reinforcement finetuning (RFT): while unconstrained RFT unlocks strong reasoning, it severely degrades trustworthiness and calibration, whereas RKL-constrained RFT preserves trustworthiness at the cost of limiting reasoning gains. To resolve this, we introduced **CARE-RFT**, a novel method that replaces the standard reverse KL penalty with a confidence-anchored, skew reverse KL divergence. This innovation provides a bounded penalty for confident, rewarded explorations while maintaining an unbounded penalty to prevent unanchored deviations. Empirically, CARE-RFT achieves a superior balance, matching the reasoning performance of unconstrained RFT while recovering the trustworthiness and calibration of the base model. Our work establishes that careful, confidence-sensitive regularization is key to building capable and trustworthy reasoning models.

## 7  ETHICS STATEMENT

This work does not involve human subjects or the release of sensitive data. We do not clearly see the harms of the applications of the proposed method either, so we are not aware of any obvious ethical concern related to this work.

## 8  REPRODUCIBILITY STATEMENT

We report all technical details for our proposed method in 4 and Appendix. The training dataset used in 5 and evaluation benchmarks 5 used in this paper is also publicly available online.

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

## A    OMITTED PROOFS

**Upper bound of the SRKL gradient coefficient.** Recall the gradient coefficient of SRKL,

$$C_\alpha(r) = \log\left(\frac{r}{\alpha r + (1-\alpha)}\right) + 1 - \alpha\,\frac{r}{\alpha r + (1-\alpha)}, \quad r = \frac{\pi_\theta}{\pi_{\text{ref}}} \geq 0$$

*Proof.* Let $d(r) = \alpha r + (1-\alpha) > 0$. Differentiating term by term,

$$\frac{\partial}{\partial r}\log\left(\frac{r}{d(r)}\right) = \frac{1}{r} - \frac{\alpha}{d(r)}, \qquad \frac{\partial}{\partial r}(1) = 0, \qquad \frac{\partial}{\partial r}\left(-\alpha\,\frac{r}{d(r)}\right) = -\alpha\,\frac{d(r) - \alpha r}{d(r)^2} = -\frac{\alpha(1-\alpha)}{d(r)^2}.$$

Hence

$$\frac{\mathrm{d}}{\mathrm{d}r}C_\alpha(r) = \frac{1}{r} - \frac{\alpha}{d(r)} - \frac{\alpha(1-\alpha)}{d(r)^2}.$$

Bring to a common denominator $r\,d(r)^2$ and simplify:

$$\frac{\mathrm{d}}{\mathrm{d}r}C_\alpha(r) = \frac{d(r)^2 - \alpha r\,d(r) - \alpha(1-\alpha)r}{r\,d(r)^2} = \frac{(\alpha r + 1 - \alpha)^2 - \alpha r(\alpha r + 1 - \alpha) - \alpha(1-\alpha)r}{r\,d(r)^2}.$$

Expanding and canceling terms,

$$(\alpha r + 1 - \alpha)^2 - \alpha r(\alpha r + 1 - \alpha) - \alpha(1-\alpha)r = (1-\alpha)^2.$$

Therefore,

$$\frac{\mathrm{d}}{\mathrm{d}r}C_\alpha(r) = \frac{(1-\alpha)^2}{r\,(\alpha r + 1 - \alpha)^2} \;\geq\; 0 \quad \text{for all } r > 0,\ \alpha \in [0,1].$$

Since $\alpha \in (0,1)$, the numerator is strictly positive, so $C_\alpha(r)$ is strictly increasing in $r$. Consequently,

$$C_\alpha(r) \leq \lim_{r\to\infty} C_\alpha(r) = \log\frac{1}{\alpha}.$$

$\square$

## B    ADDITIONAL EXPERIMENTAL RESULTS

Table 4: **Results on Qwen2.5-7B.** The trends observed on the 3B model (Table 2) are consistent at the 7B scale: CARE-RFT achieves reasoning performance comparable to unconstrained RFT while maintaining significantly better trustworthiness and calibration.

| Method | MATH | GSM8K | SelfAware | TruthfulQA | ECE $\downarrow$ |
|---|---|---|---|---|---|
| **Base Model** | 0.498 | 0.854 | 0.512 | 0.564 | 0.089 |
| GRPO (No Constraint) | 0.724 | 0.905 | 0.353 | 0.412 | 0.145 |
| RKL-GRPO | 0.602 | 0.870 | 0.491 | 0.576 | 0.095 |
| **CARE-GRPO** | **0.699** | **0.912** | **0.495** | **0.557** | **0.086** |
| DAPO (No Constraint) | 0.761 | 0.923 | 0.325 | 0.420 | 0.151 |
| RKL-DAPO | 0.641 | 0.899 | 0.487 | 0.548 | 0.090 |
| **CARE-DAPO** | **0.740** | **0.914** | **0.491** | **0.541** | **0.099** |
| GSPO (No Constraint) | 0.798 | 0.942 | 0.327 | 0.401 | 0.149 |
| RKL-GSPO | 0.673 | 0.916 | 0.498 | 0.550 | 0.088 |
| **CARE-GSPO** | **0.776** | **0.933** | **0.502** | **0.548** | **0.101** |

## C    TRAINING DETAILS

Here provides a comprehensive description of the experimental setup, including model specifications, hyperparameters, and computational environment, to ensure full reproducibility of our results.

### C.1    TRAINING CONFIGURATION

We provide a unified training configuration for all experiments. The primary differences between runs are the base model (Qwen2.5-3B or Qwen2.5-7B), the RFT algorithm (GRPO, DAPO, GSPO), and the constraint type (None, RKL, CARE-RFT). For all methods, we use the AdamW optimizer (Kingma, 2014) and a cosine learning rate scheduler with warmup.

Key hyperparameters are summarized in Table 5. The learning rate was selected via a small grid search over $\{1e\text{-}6, 3e\text{-}6, 5e\text{-}6\}$ on a held-out validation set. The $\beta$ value for the divergence penalty was set to 0.04 for all constrained methods (RKL and CARE-RFT), following common practice in prior work. For CARE-RFT, the skew parameter $\alpha$ was set to 0.8 based on the ablation study in Section 5.4.

Table 5: Summary of key hyperparameters for reinforcement finetuning.

| Hyperparameter | Value |
|---|---|
| Base Models | Qwen2.5-3B, Qwen2.5-7B (Team, 2024) |
| Training Data | MATH training set (Hendrycks et al., 2021) |
| Optimizer | AdamW |
| Learning Rate | $3 \times 10^{-6}$ |
| Learning Rate Scheduler | Cosine decay with warmup |
| Warmup Ratio | 0.03 |
| Weight Decay | 0.1 |
| Adam $\epsilon$ | $1 \times 10^{-8}$ |
| Adam $\beta_1, \beta_2$ | (0.9, 0.95) |
| Gradient Clipping | 1.0 |
| Global Batch Size | 48 (7B), 64 (3B) |
| Max Sequence Length | 2048 |
| Training Steps | 700 |
| RFT-Specific Settings | |
| Advantage Estimation | Generalized Advantage Estimation (GAE) (Schulman et al., 2017) |
| GAE $\lambda$ | 0.95 |
| Reward Normalization | varies on methods |
| Divergence Penalty $\beta$ | 0.04 (for RKL and CARE-RFT) |
| CARE-RFT Skew $\alpha$ | 0.8 |

### C.2    COMPUTATIONAL ENVIRONMENT

All experiments were conducted on a cluster of servers, each equipped with 4 NVIDIA A100 80GB GPUs. We used the VeRL framework (Sheng et al., 2024) for implementing the RFT algorithms, which is built upon PyTorch (Paszke et al., 2019). The training for each run on the Qwen2.5-3B model required approximately 6 GPU hours, while the Qwen2.5-7B model required approximately 17 GPU hours.

### C.3    REWARD FUNCTION

For all experiments on the MATH and GSM8K datasets, the reward function $r$ is defined as a binary signal indicating the final answer's correctness. Specifically, $r = 1$ if the final answer extracted from the generated response $o$ matches the ground-truth answer, and $r = -1$ otherwise. Answer extraction and matching follow the standard evaluation procedures for these benchmarks (Hendrycks et al., 2021; Cobbe et al., 2021).

## C.4 TruthfulQA and SelfAware evaluation details

**TruthfulQA.** We evaluate factual reliability and calibration on TruthfulQA (Lin et al., 2021), which consists of questions across 38 categories (health, law, finance, politics, etc.). We use the official multiple–choice (MC1) split and do not use any examples from TruthfulQA for training or hyperparameter tuning.

For each question $x$, we present the question and the answer options to the model in a single prompt and instruct it to select one option (e.g., "Answer by giving only the letter of the correct choice."). Unless otherwise noted, we decode a single completion with greedy decoding (temperature 0, top_p 1.0) and extract the model's chosen option from the first non-whitespace token. **Pass@1 accuracy** reported in Tables 2 and 4 is the fraction of questions where the chosen option matches the unique correct choice in the MC1 annotations.

For calibration analysis on TruthfulQA (ECE in Tables 2 and 4 and Figure 2), we use the same prompt but draw $K$ independent samples ($K = 10$) with temperature 0.7. For each question, we compute the empirical frequency of the most common option among the $K$ samples, $\hat{p}(x)$, and treat this as the model's confidence. The corresponding correctness label is whether the majority option is the gold answer. We then bin $(\hat{p}(x), \mathrm{correct}(x))$ pairs and compute Expected Calibration Error (ECE) using $M = 10$ equal-width confidence bins.

**SelfAware.** We evaluate models' ability to avoid hallucinating on unanswerable questions using the SelfAware benchmark introduced by (Sun et al.). In contrast to prior work that uses the full mixture of answerable and unanswerable items, our evaluation focuses exclusively on the unanswerable subset of the English QA data released by the authors, following their recommended split.

We cast each example as a single-turn QA task. For a question $x$, we prompt the model in a standard assistant style ("You are a factual assistant. If the question cannot be answered based on established facts, honestly say you do not know or that the answer is unknown.") and decode a single completion with temperature 0 and top_p 1.0. We post-process the output to extract a canonical answer string.

**Pass@1 accuracy** on SelfAware is defined as the fraction of unanswerable questions for which the model produces an explicit abstention rather than a hallucinated answer. A response is counted as correct if it expresses an abstention (e.g., contains patterns such as "I do not know", "cannot be determined", or "the answer is unknown") and does not introduce any concrete factual claim. All other responses—including partial answers or confidently stated but unsupported claims—are treated as hallucinations and scored as incorrect.

The final SelfAware score reported in Tables 2 and 4 is this abstention accuracy computed over the unanswerable subset.

## D Optimal policy under SRKL regularization

For completeness, we analyze the form of the optimal policy induced by skew reverse KL (SRKL) regularization and contrast it with the standard reverse KL (RKL) case.

**Setup.** Fix a state $s$ with action set $\mathcal{A}$, a reference policy $\pi_{\mathrm{ref}}(\cdot \mid s)$, and Q–values $Q(s, a)$ for a generic policy $\pi(\cdot \mid s)$. We consider the regularized objective

$$\max_{\pi(\cdot \mid s) \in \Delta(\mathcal{A})} \sum_{a \in \mathcal{A}} \pi(a \mid s)\, Q(s, a) \; - \; \beta\, D^{\alpha}_{\mathrm{SRKL}}\big(\pi_{\mathrm{ref}}(\cdot \mid s) \,\|\, \pi(\cdot \mid s)\big), \qquad (7)$$

where $\beta > 0$ controls regularization strength and $D^{\alpha}_{\mathrm{SRKL}}$ is the skew reverse KL divergence (**?**) used in CARE-RFT:

$$D^{\alpha}_{\mathrm{SRKL}}\big(\pi_{\mathrm{ref}}(\cdot \mid s) \,\|\, \pi(\cdot \mid s)\big) = \sum_{a \in \mathcal{A}} \pi(a \mid s) \log \frac{\pi(a \mid s)}{\alpha\, \pi(a \mid s) + (1 - \alpha)\, \pi_{\mathrm{ref}}(a \mid s)}, \quad \alpha \in (0, 1).$$
$$(8)$$

This is the same divergence as Eq. (5) in the main text, written at the state–action level.[1]

---

[1]For brevity we suppress the dependence on $s$ in what follows.

**Reparameterization by likelihood ratios.**   It is convenient to express everything in terms of the likelihood ratio

$$r(a) \; \frac{\pi(a \mid s)}{\pi_{\mathrm{ref}}(a \mid s)}, \qquad \pi(a \mid s) = r(a)\,\pi_{\mathrm{ref}}(a \mid s), \tag{9}$$

for all $a$ such that $\pi_{\mathrm{ref}}(a \mid s) > 0$. The normalization constraint $\sum_a \pi(a \mid s) = 1$ becomes

$$\sum_{a \in \mathcal{A}} \pi_{\mathrm{ref}}(a \mid s)\, r(a) \;=\; 1. \tag{10}$$

In terms of $r$, the SRKL divergence can be written as an expectation under $\pi_{\mathrm{ref}}$:

$$D_{\mathrm{SRKL}}^{\alpha}\big(\pi_{\mathrm{ref}} \| \pi\big) = \sum_{a \in \mathcal{A}} \pi_{\mathrm{ref}}(a \mid s)\, r(a) \left[\log r(a) - \log\big(\alpha\, r(a) + 1 - \alpha\big)\right]. \tag{11}$$

Using equation 11, the per-state regularized objective equation 7 becomes

$$\mathcal{L}(r, \lambda) \sum_a \pi_{\mathrm{ref}}(a \mid s)\, r(a)\, Q(s,a) - \beta \sum_a \pi_{\mathrm{ref}}(a \mid s)\, r(a) \left[\log r(a) - \log\big(\alpha r(a) + 1 - \alpha\big)\right]$$

$$+ \lambda\Big(1 - \sum_a \pi_{\mathrm{ref}}(a \mid s)\, r(a)\Big), \tag{12}$$

where $\lambda$ is the Lagrange multiplier enforcing the normalization constraint equation 10.

**First-order optimality condition.**   Differentiating equation 12 with respect to $r(a)$ and using the chain rule, we use the fact (cf. Eq. (6)–(7) in the main text) that

$$\frac{\partial}{\partial r} D_{\mathrm{SRKL}}^{\alpha}\big(\pi_{\mathrm{ref}} \| \pi\big) = \pi_{\mathrm{ref}}(a \mid s)\, C_{\alpha}\big(r(a)\big), \tag{13}$$

where the SRKL gradient coefficient is

$$C_{\alpha}(r) = \log \frac{r}{\alpha r + 1 - \alpha} + 1 - \frac{\alpha r}{\alpha r + 1 - \alpha}, \qquad r > 0,\ \alpha \in (0,1). \tag{14}$$

Therefore

$$\frac{1}{\pi_{\mathrm{ref}}(a \mid s)} \frac{\partial \mathcal{L}}{\partial r(a)} = Q(s,a) - \beta\, C_{\alpha}\big(r(a)\big) - \lambda. \tag{15}$$

At an interior optimum (i.e., for actions $a$ with $\pi^*(a \mid s) > 0$), the stationarity condition $\partial \mathcal{L}/\partial r(a) = 0$ yields

$$Q(s,a) - \beta\, C_{\alpha}\big(r^*(a)\big) = \lambda(s), \qquad r^*(a) = \frac{\pi^*(a \mid s)}{\pi_{\mathrm{ref}}(a \mid s)}. \tag{16}$$

Equivalently,

$$Q(s,a) - \lambda(s) = \beta\, C_{\alpha}\left(\frac{\pi^*(a \mid s)}{\pi_{\mathrm{ref}}(a \mid s)}\right). \tag{17}$$

This characterizes the optimal policy under SRKL regularization: for each state $s$, the optimal likelihood ratio $r^*(a) = \pi^*(a \mid s)/\pi_{\mathrm{ref}}(a \mid s)$ is the unique solution of equation 17 consistent with the normalization constraint equation 10.

**Monotonicity and boundedness.**   Appendix A shows that $C_{\alpha}(r)$ is strictly increasing in $r$ and admits a finite upper bound (**?**, see also Eq. (7)):

$$\frac{d}{dr} C_{\alpha}(r) = \frac{(1-\alpha)^2}{r\big(\alpha r + 1 - \alpha\big)^2} > 0, \qquad \lim_{r \to \infty} C_{\alpha}(r) = \log \frac{1}{\alpha}, \qquad \lim_{r \to 0} C_{\alpha}(r) = -\infty. \tag{18}$$

As a consequence, for fixed $\lambda(s)$:

- *Monotone reweighting.* From equation 17 and the strict monotonicity of $C_{\alpha}$, we have

$$Q(s, a_1) > Q(s, a_2) \iff r^*(a_1) > r^*(a_2).$$

  That is, SRKL does not change the ordering of actions by $Q(s,a)$; it reweights $\pi_{\mathrm{ref}}$ in a monotone way in the Q–values.

- *Softly bounded upward deviations.* Since $C_\alpha(r) \leq \log(1/\alpha)$ for all $r > 0$, the effective advantage $Q(s, a) - \lambda(s)$ realized at the optimum obeys

$$Q(s, a) - \lambda(s) = \beta \, C_\alpha\big(r^*(a)\big) \leq \beta \log \frac{1}{\alpha}. \tag{19}$$

  Thus, for actions whose Q–values are much larger than the state-dependent baseline $\lambda(s)$, the corresponding $r^*(a)$ lies in a regime where $C_\alpha$ is close to its finite ceiling $\log(1/\alpha)$, and further increases in $Q(s, a)$ only have a diminishing effect on the likelihood ratio. This induces a *soft clipping* of upward deviations from $\pi_{\text{ref}}$, consistent with the bounded penalty discussed in Section 4.

- *Unbounded penalty on downward deviations.* As $r \to 0$, $C_\alpha(r) \to -\infty$ (Eq. equation 18), so actions with sufficiently low Q–values relative to $\lambda(s)$ can only satisfy equation 17 with extremely small ratios $r^*(a) \ll 1$. In other words, SRKL still imposes an effectively unbounded penalty on *downward* deviations from the reference policy, strongly discouraging the model from deleting probability mass on tokens that $\pi_{\text{ref}}$ considers likely.

Taken together, equation 17–equation 18 formalize the asymmetric behavior described in the main text: SRKL induces a confidence-sensitive regularization that (i) allows high-Q actions to become more probable than under $\pi_{\text{ref}}$ without letting the likelihood ratio explode, while (ii) preserving strong anchoring against unwarranted probability decreases. This explains why CARE-RFT can maintain the calibration of the base model while still enabling the exploratory shifts needed for improved reasoning.

**Connection to the RKL case.** For comparison, when $\alpha \to 0$, SRKL reduces to RKL and the gradient coefficient becomes $C_0(r) = \log r + 1$. The first-order condition equation 17 specializes to

$$Q(s, a) - \lambda(s) = \beta\big(\log r^*(a) + 1\big) \quad \Longleftrightarrow \quad r^*(a) \propto \exp\!\Big(\tfrac{1}{\beta} Q(s, a)\Big), \tag{20}$$

so that the optimal policy takes the familiar exponential-tilting form

$$\pi^*(a \mid s) \;\propto\; \pi_{\text{ref}}(a \mid s) \, \exp\!\Big(\tfrac{1}{\beta} Q(s, a)\Big), \tag{21}$$

with unbounded log-ratios $\log \frac{\pi^*(a|s)}{\pi_{\text{ref}}(a|s)}$. In contrast, under SRKL with $\alpha > 0$, the mapping from Q–values to likelihood ratios is given implicitly by equation 17 through the bounded, strictly increasing $C_\alpha$, which tempers these deviations and yields the confidence-anchored behavior exploited by CARE-RFT.

# E   RELATED WORKS

**Reinforcement Finetuning.** Reinforcement finetuning (RFT) has become the central paradigm for scaling reasoning in large language models. Early approaches relied on PPO-based RLHF with explicit reward models (Ouyang et al., 2022), while recent work such as GRPO (Shao et al., 2024) demonstrated that critic-free updates are sufficient to elicit strong reasoning behaviors in long chain-of-thought tasks. This simplicity has fueled a rapid proliferation of variants targeting GRPO's efficiency and stability limits. For example, DAPO (Yu et al., 2025) modifies group-normalized advantages and removes KL constraints to accelerate divergence from the base model; GMPO (Zhao et al., 2025) replaces GRPO's arithmetic mean reward aggregation with a geometric mean for better stability; Dr.GRPO (Liu et al., 2025) restores an unbiased policy gradient objective by removing the length and std normalization terms; and GSPO (Zheng et al., 2025) explicitly defines the importance ratio based on sequence likelihood rather than per-token likelihood, and it applies clipping, rewarding, and optimization at the sequence level. Collectively, these works show the field's push toward more aggressive policy optimization in pursuit of stronger reasoning. Yet this trend has unintended consequences. Removing divergence constraints amplifies entropy collapse, driving models toward overconfident predictions that erode calibration and factual reliability. At the same time, most existing RFT methods propagate a single response-level advantage uniformly to all tokens in the Chain-of-Thought (CoT). As a result, once a final answer is deemed correct, every intermediate step is reinforced equally—even if some steps are logically unsound or factually spurious. This

coarse-grained credit assignment not only distorts the learning signal for reasoning but also encourages the persistence of hallucinated intermediate content, making errors more systematic and harder to detect in long-CoT training.

**Hallucination and Calibration in RFT.** At the same time, an emerging line of work highlights a concerning byproduct of RFT: reasoning hallucinations. Unlike surface-level errors, these occur when models produce logically coherent but factually incorrect reasoning traces (Sun et al., 2025; Lanham et al., 2023). Process-supervision methods attempt to mitigate this by providing step-wise rewards (Lightman et al., 2023; Wang et al., 2023), either through human-labeled steps (Jiao et al., 2024; Lin et al., 2025) or model-generated signals (Xu et al., 2025; Mitra & Ulukus, 2025; Wang et al., 2025; Li et al., 2025). Inference-time verification offers another avenue: approaches such as Chain-of-Verification (Dhuliawala et al., 2023), Ever (Kang et al., 2023), or HalluciNot (Paudel et al., 2025) validate outputs dynamically or post-hoc to reduce hallucination risk. While effective, these solutions introduce significant annotation or inference overhead and do not directly address the root cause of miscalibration during training. Indeed, recent studies find that RFT without KL regularization tends to sharpen token distributions indiscriminately, yielding overconfident predictions on ambiguous or fact-seeking queries (Song et al., 2025; Yao et al., 2025). This gap motivates a closer look at how divergence constraints influence both reasoning ability and trustworthiness.

**Divergence-Regularized Policy Optimization.** Classical policy optimization stabilizes learning with KL regularization (Schulman et al., 2017; 2015), grounding updates against a reference model to prevent reward hacking and overoptimization. Yet in long-CoT settings, reverse KL can be overly restrictive(Yu et al., 2025), discouraging the exploratory deviations needed for emergent reasoning. This tension has driven many GRPO variants to remove KL entirely (Yu et al., 2025; Liu et al., 2025; Zheng et al., 2025; Zhao et al., 2025), thereby amplifying the miscalibration problem. Beyond standard KL, prior work in machine learning has explored divergences—such as Jensen–Shannon (JS) divergence (Zawar et al.) from the family of f-divergences (Belousov & Peters, 2017). However, JS divergence requires sampling from the old policy where with standard KL we only need data from the new policy and log-probabilities of the old policy at those sampled actions. However, estimating JS divergence will complicate each update with samples from reference policy. Other directions, such as $\alpha$-divergences and Rényi divergences, offer tunable tradeoffs between mode-seeking and mode-covering behavior (Li & Turner, 2016; Minka et al., 2005), but suffer from either intractable estimation or unstable gradients when applied in token-level policy optimization. More importantly, none of these alternatives directly address the calibration collapse observed in unconstrained RFT: they either over-penalize exploration, limiting reasoning gains, or fail to prevent indiscriminate sharpening that drives overconfidence and hallucination. This gap highlights the need for a divergence measure that (i) is tractable under on-policy sampling, (ii) provides stable gradients that preserve moderate entropy rather than collapsing it, and (iii) adapts penalties asymmetrically to the model's confidence, discouraging unjustified certainty while still allowing high confidence deviations when warranted.

