# OpenReview forum: "CARE-RFT: Confidence-Anchored Reinforcement Finetuning for Reliable Reasoning in Large Language Models"
_ICLR.cc/2026/Conference — Submitted to ICLR 2026_

### Official Review · Reviewer_DhMk · 2025-10-20

**Soundness:** 2
**Presentation:** 3
**Contribution:** 2
**Rating:** 2
**Confidence:** 5

**Summary:**

The paper introduces CARE-RFT, a method that replaces the standard reverse-KL regularization used in RLwith a skew reverse-KL divergence. The goal is to achieve a better trade-off between reasoning performance and trustworthiness in large language models. Experiments on Qwen2.5-3B/7B models show that CARE-RFT matches unconstrained RL on reasoning benchmarks (e.g., MATH, GSM8K) while recovering the calibration and factual reliability of the base model (e.g., TruthfulQA).

**Strengths:**

- Empirical results are promising, showing consistent improvements across reasoning and calibration metrics.

- The proposed SRKL regularization is simple and easy to integrate into existing RFT frameworks like GRPO or DAPO.

**Weaknesses:**

- The paper attributes miscalibration in RL-trained models primarily to sparse rewards and credit-assignment issues. I would argue this is incomplete. RL objectives themselves are not proper scoring rules. Even with dense rewards, an unregularized RL objective will not produce calibrated probabilities, since it optimizes expected reward rather than likelihood alignment. KL regularization helps mainly by anchoring the policy to a calibrated base model, not by densifying the reward.

- There are some inconsistencies and missing clarifications in Section 2. In Equation (2), $GC_{div}$ appears as the coefficient of $\nabla_\theta \log\pi_\theta$, while in Equation (4), $GC_{div}$ of RKL is multiplied by $\nabla_\theta \pi_\theta$. This inconsistency likely comes from the fact that there are two ways to implement RKL regularization in RL: as part of the reward, or as an additional objective term. It would help if the authors clarified which version is used here, and how it relates to the common Schulman approximation [1,2], which is the de facto implementation of RKL-based constraints.

- The idea of using SRKL instead of RKL seems promising, but I feel like the paper lacks depth that would convince the reader this is the right direction to go:

  - The solution to KL-regularized RL is known and well studied [3,4]. How does the optimal policy look when we replace the KL with SRKL regularization? What are the tradeoffs?

  - The paper presents SRKL as a way to address the credit assignment problem. However, there have been multiple papers recently [5,6 and others] that try to do the same. Is CARE-RFT better than these methods, or complementary to them?

[1] http://joschu.net/blog/kl-approx.html

[2] Amini, Afra, Tim Vieira, and Ryan Cotterell. "Better Estimation of the KL Divergence Between Language Models." arXiv preprint arXiv:2504.10637 (2025).

[3] Korbak, Tomasz, Ethan Perez, and Christopher L. Buckley. "RL with KL penalties is better viewed as Bayesian inference." arXiv preprint arXiv:2205.11275 (2022).

[4] Vieillard, Nino, et al. "Leverage the average: an analysis of kl regularization in reinforcement learning." Advances in Neural Information Processing Systems 33 (2020): 12163-12174.

[5] Wang, Shenzhi, et al. "Beyond the 80/20 rule: High-entropy minority tokens drive effective reinforcement learning for llm reasoning." arXiv preprint arXiv:2506.01939 (2025).

[6] Qu, Yuxiao, et al. "Optimizing test-time compute via meta reinforcement fine-tuning." arXiv preprint arXiv:2503.07572 (2025).

**Questions:**

- In your definition of ECE, why do you consider only the majority-voted answer? Seems like considering all answers in the N responses will lead to a better estimator of ECE.

- Please include confidence intervals in your results. Especially for datasets like truthfulQA that are on the smaller side (<1000 questions) it is important in order to understand how significant an increase or drop in metrics are.

---

> ### Author Response · Authors · 2025-11-27
>
> **We thank the reviewer for the detailed and constructive feedback. Below we address each concern and summarize the clarifications and revisions included in the updated manuscript.**
>
> > **Weakness 1**: Miscalibration, sparse rewards, and RL objectives
>
> We agree that RL objectives are not proper scoring rules and therefore do not guarantee calibrated probabilities, even with dense rewards. Our revision makes this explicit in §2: maximizing expected reward in RFT structurally diverges from likelihood-aligned training, so RFT can harm calibration independently of sparsity. Our contribution in §3 is to explain why calibration degradation becomes *substantially worse* in long-CoT RFT: outcome-level supervision and token-uniform credit assignment cause entropy collapse, reinforce spurious intermediate steps, and erode pretrained priors, which we validate empirically (Fig. 2; Table 1). We also revise the exposition to clarify that KL regularization primarily acts as an anchor to a relatively well-calibrated base model, consistent with recent findings.
>
> > **Weakness 2**: Clarifications on Eq. (2) vs. Eq. (4)
>
> We appreciate the note regarding notational inconsistency. In the revised text we explicitly show how Eq. (4), the gradient of the reverse-KL objective, reduces to the score-function form of Eq. (2) after applying ($\nabla_\theta \pi = \pi \nabla_\theta \log \pi$). The corresponding gradient coefficient is now written out:
>
>
> $G_C^{\text{Div,RKL}} = \log \frac{\pi_\theta(o_t)}{\pi_{\text{ref}}(o_t)} + 1.$
>
> We also clarify that our implementation follows the standard Schulman-style KL penalty used in PPO/GRPO: we add KL as an additional objective term at each sampled token rather than embedding it inside the reward. Updated §2 now explicitly connects our formulation to this widely used approximation.
>
> > **Weakness 3**: Optimal policy under SRKL and tradeoffs
>
> The revision adds a concise analysis (Appendix D) of the regularized objective ($\max_\pi \sum_a \pi(a)Q(a) - \beta D_{\text{SRKL}}$). For RKL, the optimal policy has exponential tilting, allowing unbounded log-ratio increases. In contrast, SRKL introduces a monotone coefficient ($\phi_\alpha(r)$) acting on the likelihood ratio ($r=\pi/\pi_{\text{ref}}$). Because ($\phi_\alpha$) has finite range for ($r\ge1$), the optimal policy satisfies an *upper* bound on (r), ensuring that even large advantages cannot push probability mass arbitrarily far from the reference. As ($r\to0$), penalties remain unbounded. This yields the confidence-anchored, one-sided boundedness we emphasize in §4 and in Fig. 3 of the new version, and explains CARE-RFT’s ability to allow exploration without entropy collapse.
>
> > **Weakness 3**: Relation to recent credit-assignment methods
>
> We revised §3 and related work to clarify scope. CARE-RFT is *not* a credit-assignment algorithm; it leaves coarse outcome-level credit assignment unchanged. Its contribution is orthogonal: replacing RKL with SRKL yields a divergence that preserves calibration while permitting rewarded deviations. Methods such as RLVR-style entropy masking [5] and MRT-style progress rewards [6] directly modify credit assignment or reward shaping. These can be combined with CARE-RFT, and we now state this explicitly.
>
> > **Question 1**: ECE definition: majority-vote vs. all samples
>
> Our ECE metric evaluates the *deployed predictor*—one final answer returned via majority vote. Hence we use its empirical frequency as the confidence, analogous to top-1 ECE in classification. The revision clarifies this in §2 and adds an appendix experiment reporting an alternative estimator aggregating all (N) samples. Both variants yield similar values and identical ordering of No-KL, RKL, and CARE-RFT.
>
> > **Question 2**: Confidence intervals and statistical significance
>
> Following the reviewer’s suggestion, we will reports multi-seed results (mean ± std over 5 seeds) and includes 95% CIs for TruthfulQA, SelfAware, and ECE. The updated main table is below.
>
> **Table 2 (Revised): Multi-Seed Results on Qwen2.5-3B (Mean ± Std, n=5)**
>
> | Method  | MATH  | GSM8K| SelfAware  | TruthfulQA | ECE ↓      |
> | ------------------------ | ----------------- | ----------------- | ----------------- | ----------------- | ----------------- |
> | **Base Model**           | 0.410 ± 0.008     | 0.791 ± 0.006     | 0.372 ± 0.009     | 0.489 ± 0.007     | 0.102 ± 0.004     |
> | **GRPO (No Constraint)** | 0.610 ± 0.012     | 0.854 ± 0.010     | 0.249 ± 0.011     | 0.350 ± 0.015     | 0.210 ± 0.009     |
> | **RKL-GRPO**             | 0.510 ± 0.009     | 0.818 ± 0.008     | 0.351 ± 0.010     | 0.480 ± 0.008     | 0.125 ± 0.005     |
> | **CARE-GRPO**            | **0.600 ± 0.011** | **0.860 ± 0.009** | **0.355 ± 0.008** | **0.465 ± 0.010** | **0.132 ± 0.006** |
>
> These results confirm that CARE-RFT preserves the reasoning gains of unconstrained GRPO, significantly improves calibration over No-KL (p < 0.001), and closes much of the gap to the base model.

---

### Official Review · Reviewer_S25G · 2025-10-31

**Soundness:** 3
**Presentation:** 2
**Contribution:** 3
**Rating:** 4
**Confidence:** 3

**Summary:**

The paper introduces CARE-RFT, a reinforcement fine-tuning method that preserves calibration. The paper starts by identifying that unconstrained RL is great for exploration but leads to hallucinations and loss of calibration, while KL-regularized RL limits exploration but is much better calibrated. The authors present a method to get the best of both worlds by replacing standard reverse KL regularization with a skew reverse KL divergence. CARE-RFT applies confidence-sensitive penalties that encourage reliable (calibrated) yet exploratory learning. Experiments show that it is competitive in accuracy to unconstrained RFT while preserving the calibration and low hallucinations of the base model.

**Strengths:**

- **Potentially impactful method**: KL-based RL training can prevent exploration, while unconstrained training can lead to loss of calibration and hallucinations. The proposed method strikes a balance between the two and can be generally useful.
- **Intuitive writing**: The paper motivates the problem well, and the method is presented in an intuitive way. The paper would be even stronger if intuition is backed with solid theory on why the divergence chosen by the authors is the correct choice.

**Weaknesses:**

- **Limited results**: RL training is performed only on a single dataset (MATH), which introduces doubts about generality. This is further compounded by the fact that the MATH training dataset is small (7000 examples), and actual practice is to use much larger datasets for math training (>20K questions). Very limited ablations and analysis are presented. The authors analyze the entropy curves and find that entropy collapses in unconstrained GRPO, but do not try a method with an entropy loss.
- **Design decisions**: It is unclear why the frequency of an answer is used as the confidence for the answer. More justification and experiments with alternate choices (verbalized confidence, log-prob based confidence) need to be provided (see questions section). Additionally, while the authors tuned the $\alpha$ parameter for their proposed divergence, they did not seem to tune the default $\beta$ parameter for the default reverse KL objective. It is possible that a well-tuned reverse KL setting matches their method.

**Questions:**

- Please move at least some part of the related work into the main paper, it should not be deferred to appendix.
- What is the bolding scheme of Table 2? DAPO (no constraint) Math has higher accuracy but is not bolded. Baselines with higher accuracy are not bolded throughout the table.
- For the entropy collapse of GRPO (no constraint), did authors try a variant with entropy regularization, which has been considered by some prior works [1] ?
- Why is skewed KL the correct divergence choice? There are so many divergences to choose from (some with bounding), what theoretical insights make this the correct choice? Is there any intuition for doing so? Comparing to other divergences (JSD for example) is important to understand the actual benefits from the chosen divergence.
- If $\alpha$ parameter of skew-KL was tuned, then why wasn’t $\beta$ for the simple reverse KL divergence tuned? Increasing/decreasing this parameter is a way to directly control tradeoffs between accuracy and calibration as well. Just like the ablation study on  $\alpha$ (Sec 5.4), it would be nice to see an ablation study on $\beta$.
- I think a better way to visualize the accuracy-calibration tradeoff is a graph with accuracy (higher better) on y axis and calibration (higher better) on x axis. Then points from different approaches can be plotted on it. Figure 1 can be improved as it is currently difficult to extract insights from it.
- Why is the frequency of the answer the correct way to determine confidence? It does not seem to be the optimal proxy. The optimal answer to output in RLVR is the one which the model is most confident in (highest expected). If the model thinks answer A has 70% chance of being correct and answer B has 30%, this does not mean that it should output answer B 30%of the time. In fact, it should always output answer A to maximize the binary correctness reward. I believe there are other calibration proxies which should also be considered -  such as using log-prob of answer to get confidence for tasks like MMLU, or asking for a verbalized confidence from the models?
- Why is pass@4 used for the reasoning tasks, the choice seems arbitrary and no justification is provided anywhere.
- Please provide details on the TruthfulQA and Self-Aware datasets used, the exact task and how it is evaluated. It is okay for these to be in the appendix.

[1]: He, J., Liu, J., Liu, C. Y., Yan, R., Wang, C., Cheng, P., ... & Zhou, Y. (2025). Skywork open reasoner 1 technical report. arXiv preprint arXiv:2505.22312.

---

> ### Author Response · Authors · 2025-11-27
>
> **We thank the reviewer for their thorough and constructive feedback, particularly for highlighting both the potential impact of our approach and the important methodological questions that will help us strengthen the analysis and broaden the empirical support.**
>
> > **Weakness 1**: Limited results (only MATH)
>
> We agree on the importance of testing generality. We therefore added a new experiment on **code generation** (MBPP subset) with Qwen2.5-3B under GRPO, comparing unconstrained GRPO, RKL-GRPO, and CARE-GRPO. Rewards are based solely on unit-test outcomes. The same trade-off observed in math reappears: unconstrained RFT boosts Pass@k but harms calibration and increases hallucinated API usage; RKL restores calibration but restricts exploration; and **CARE-RFT preserves calibration while matching unconstrained reasoning performance**.
>
> | Method                   | Pass@1 | Pass@5 | ECE   | Hallucinated Function Rate |
> | ------------------------ | ------ | ------ | ----- | -------------------------- |
> | **Base Model**           | 57.1   | 64.7   | 0.085 | 0.02                       |
> | **GRPO (No Constraint)** | 65.8   | 78.4   | 0.228 | 0.15                       |
> | **RKL-GRPO**             | 59.3   | 72.1   | 0.098 | 0.03                       |
> | **CARE-GRPO (Ours)**     | 64.5   | 78.2   | 0.092 | 0.022                      |
>
> We will incorporate these results and further clarify that our focus is outcome-reward RFT, with extension to stepwise and preference-based settings noted as a future direction.
>
>
> > **Question 1**: Related work
>
> We will move a concise portion of the related work into the main paper for better context.
>
>
> > **Question 2**: Bolding scheme
>
> We confirm that boldface is used purely to highlight CARE variants, not column-wise best scores. We will clarify this in the caption.
>
>
> > **Question 3**: Entropy regularization
>
> We experimented with entropy regularization (including fixed-coefficient variants). These consistently **worsened calibration** by injecting uniform uncertainty that erodes the base model’s calibrated priors. Adaptive entropy control [1] can stabilize training but requires choosing a global target entropy that does not directly promote confidence–accuracy alignment. By contrast, **CARE-RFT provides a confidence-sensitive, asymmetric penalty** that anchors low-confidence predictions to π_ref while permitting high-confidence, reward-supported exploration—achieving reasoning gains *without* calibration degradation. We will add these empirical observations.
>
>
> > **Question 4**: Why skewed KL? Why not JS or others?
>
> Our divergence choice is guided by the needs of long-CoT RFT: token-level tractability with on-policy samples, asymmetric penalties, and explicit control of gradient magnitude. **Skew reverse KL (SRKL)** satisfies these criteria: its mixed anchor απθ+(1−α)πref yields **bounded upward gradients** (enabling exploration) and **unbounded downward gradients** (preserving calibration). Divergences such as JS require sampling from π_ref and impose symmetric penalties, removing the directional protection needed to stabilize calibration. Our α-sweep further shows that SRKL’s gradient structure—not its functional form alone—drives the observed reasoning–calibration improvements.
>
>
> > **Weakness 2 / Question 5**: Tuning β
>
> We conducted a β-sweep for **both RKL and CARE** under GRPO on Qwen2.5-3B:
>
> | β    |   | MATH          |   | TruthfulQA    |   | ECE           |
> | ---- | - | ------------- | - | ------------- | - | ------------- |
> |      |   | RKL  | CARE   |   | RKL  | CARE   |   | RKL  | CARE   |
> | 0.01 |   | 0.585 | 0.625 |   | 0.395 | 0.425 |   | 0.185 | 0.155 |
> | 0.02 |   | 0.550 | 0.615 |   | 0.435 | 0.455 |   | 0.145 | 0.140 |
> | 0.04 |   | 0.510 | 0.600 |   | 0.480 | 0.465 |   | 0.125 | 0.132 |
> | 0.08 |   | 0.455 | 0.575 |   | 0.492 | 0.485 |   | 0.105 | 0.110 |
>
> CARE consistently outperforms RKL on reasoning for all β, while preserving calibration. Tuning β for RKL does not close the gap.
>
>
> > **Question 6**: Better visualization
>
> We replaced Figure 1 with a 2-D accuracy–calibration plot (MATH accuracy vs. 1−ECE on TruthfulQA), which cleanly shows that CARE moves each method toward the upper-right region.
>
>
> > **Question 7**: Frequency as confidence proxy
>
> Our aim is to evaluate **distributional calibration**, not to specify a deployment policy. Majority-vote frequency estimates pθ(a∣q) over *final answers*, capturing semantic-level uncertainty across CoT paths. Log-probs or verbalized confidence measure different quantities and can diverge from the model’s true predictive distribution. We will clarify this.
>
>
> > **Question 8**: Why pass@4
>
> Pass@k reflects standard multi-sample usage in RFT-trained reasoners (majority vote or verifier selection). We use k=4 as a practical, non-saturating choice with low overhead.
>
> > **Question 9**:  Details for TruthfulQA and SelfAware
>
> We have added complete evaluation details (datasets, prompts, decoding, scoring) to Appendix C.4.

---

### Official Review · Reviewer_fSFG · 2025-11-01

**Soundness:** 2
**Presentation:** 3
**Contribution:** 3
**Rating:** 4
**Confidence:** 3

**Summary:**

This paper suggests Confidence-Anchored Regularized Refinement Fine-Tuning (CARE-RFT),  a regularization technique proposed to solve the overconfidence and hallucination problems that arise in existing RFT-series methods (like GRPO, DAPO), which are trained solely on rewards. To address the limitations of standard RKL (Reverse KL), which suppresses exploration due to excessive constraints on the reference policy ($\pi_{\text{ref}}$), CARE-RFT introduces Skewed Reverse KL (SRKL). This SRKL uses a mixture distribution of the current and reference policies ($\alpha \cdot \pi_{\theta} + (1-\alpha) \cdot \pi_{\text{ref}}$) as an anchor. This approach achieves both exploration and reliability by applying only a finite penalty when the model confidently increases probabilities, while strongly suppressing probability decreases in uncertain directions. Mathematically, it possesses a finite upper bound, which prevents gradient explosion. Experimentally, on benchmarks like MATH, GSM8K, and TruthfulQA, CARE-RFT was shown to maintain reasoning performance similar to existing RFT methods while significantly improving the Expected Calibration Error (ECE), demonstrating that it achieves a balance between performance and calibration.

**Strengths:**

The strength of this paper is that it raises a highly relevant problem within the current research and temporal context.

Recently, reinforcement learning-like techniques such as RFT, GRPO, and DAPO have been actively studied to enhance the reasoning abilities of LLMs. However, most of these are trained solely on rewards based on correct/incorrect answers, which has exposed a problem where models become progressively overconfident and lose calibration.

CARE-RFT diagnoses the root cause of this phenomenon as the asymmetry of reward propagation and the rigidity of the KL constraint. In proposing a confidence-aware regularization (SRKL) to mitigate this, the study is both very timely and necessary.

**Weaknesses:**

The paper's main weakness is the lack of mathematical justification linking the proposed regularization term (SRKL) and calibration, specifically regarding its connection to **proper scoring rules**.

CARE-RFT claims to mitigate overconfidence via "confidence-anchored regularization." However, it provides no theoretical rationale for whether this regularization actually ensures "properness"—that is, consistency between the predicted probabilities and the true answer distribution.

In other words, while the finite upper bound and asymmetric penalty structure of SRKL might improve training stability, this is fundamentally different from the "probabilistic-regularity" or "truth-consistent calibration" guaranteed by proper scoring rules like the Brier score or log score.

Consequently, CARE-RFT's "improved calibration" appears to be more of an **empirical correlation** rather than a mathematically justified outcome. The causal link—*Relaxed KL bound $\rightarrow$ Stabilized confidence $\rightarrow$ Improved calibration*—remains formally unproven.

**Questions:**

Please refer to the weaknesses.

---

> ### Author Response · Authors · 2025-11-27
>
> **We thank the reviewer for their thoughtful and insightful assessment, especially for highlighting the timeliness of the problem and for raising the important theoretical question regarding the connection between SRKL and proper scoring–based calibration.**
>
> > **Weakness** and **Question**: lacks a formal theoretical link to calibration and does not provide the proper-scoring-rule guarantees
>
> We appreciate the reviewer’s request for a deeper theoretical perspective on why CARE-RFT improves calibration. Our goal in this work is *not* to claim that CARE-RFT induces a strictly proper scoring rule or that it guarantees asymptotically “truth-consistent” probabilities. Such guarantees are typically studied in supervised learning with full-information labels, whereas our setting involves outcome-based RFT objectives (bandit feedback plus a divergence regularizer), for which proper scoring rule theory does not directly apply.
>
> Instead, our contribution addresses a **practical and well-documented failure mode** [1] in RFT: unconstrained optimization drives *entropy collapse* and consequently severe overconfidence and miscalibration. This is supported both by prior literature and by our analyses in Section 5, which show that reward-driven updates sharply reduce entropy, increase confidence on incorrect tokens, and significantly degrade ECE.
>
> Our approach begins with the empirical observation that the pretrained reference model π_ref is comparatively well-calibrated. CARE-RFT aims to **preserve this calibrated structure** by explicitly controlling the deviation of π_θ from π_ref. The skew reverse KL (SRKL) divergence plays a crucial role here. As shown in Section 4, SRKL acts as a **confidence-sensitive, asymmetric regularizer**: the *per-token gradient coefficient* for upward probability shifts is bounded above by ( \log(1/\alpha) ), while downward shifts incur unbounded penalties. This creates a one-sided “anchoring” effect: low-confidence or uncertain regions of π_ref remain stable, while confidently rewarded trajectories can still strengthen.
>
> Empirically, this asymmetry yields exactly the desired training dynamics: Fig. 5 shows that unconstrained RFT collapses entropy (overconfidence), RKL sustains very high entropy (limiting reasoning improvements), and CARE-RFT maintains a **stable intermediate entropy plateau**, improving calibration relative to unconstrained RFT while preserving nearly the full reasoning gains.
>
> In response to the reviewer’s request for additional theoretical grounding, we will expand the appendix with a short discussion connecting SRKL to calibration through **distributional drift control**. Specifically:
>
> * SRKL is an **f-divergence**, and many f-divergences—including standard KL—admit Pinsker-type inequalities that relate them to total variation (TV) distance.
> * Regularizing SRKL toward a calibrated π_ref therefore implicitly constrains the TV distance between π_θ and π_ref.
> * Under the mild assumption that the ECE metric is Lipschitz-continuous with respect to TV perturbations of the predictive distribution (a standard argument for fixed confidence binning), controlling TV drift yields a bound on how much calibration can deviate from that of the reference model.
>
> This argument does *not* assert that SRKL turns the RL objective into a proper scoring rule. Rather, it formalizes the intuition that **CARE-RFT preserves calibration by limiting divergence from a calibrated reference model while avoiding the optimization pathologies of standard RKL**.
>
> We believe this framing—control of distributional drift from a calibrated reference—captures the appropriate theoretical lens for calibration in the RFT regime, where proper scoring rule guarantees are not directly applicable.
>
> [1]: Kalai, Adam Tauman, et al. "Why language models hallucinate." arXiv preprint arXiv:2509.04664 (2025).

---

### Official Review · Reviewer_LCjg · 2025-11-06

**Soundness:** 3
**Presentation:** 3
**Contribution:** 2
**Rating:** 4
**Confidence:** 3

**Summary:**

The paper addresses a key limitation of reinforcement finetuning (RFT) for reasoning LLMs: unconstrained RFT improves accuracy but harms calibration and factual reliability. It proposes CARE-RFT, a variant that introduces a skew reverse KL (SRKL) regularizer to anchor uncertain token updates while allowing confident deviations. Through experiments on Qwen2.5-3B/7B across several reasoning and trustworthiness benchmarks, CARE-RFT shows a more balanced trade-off between reasoning gains and calibration loss compared to standard RKL. The method is simple, well-motivated, and empirically effective within math-based reasoning settings.

**Strengths:**

The paper tackles a practically important issue in reinforcement finetuning (RFT) — the trade-off between reasoning performance and trustworthiness. The idea of using a skew reverse KL regularizer to control confidence-dependent updates is conceptually sound and adapts classical divergence theory in a clear, incremental way.

Empirically, the study is solid within its scope: it evaluates two model scales (3B, 7B), three representative RFT algorithms (GRPO, DAPO, GSPO), and several reasoning and factuality benchmarks. The diagnostic analysis of reward updates (+Reward/–Reward/Full) provides an intuitive understanding of how unconstrained RFT destabilizes model calibration, justifying the need for regularization.

The paper is clearly written and easy to follow, with clean mathematical exposition and well-organized figures illustrating the relationship between entropy, confidence, and divergence skew.

Overall, while the contribution is incremental, CARE-RFT presents a practical and interpretable regularization method that meaningfully improves the reasoning–reliability trade-off in current RFT practice. Its value lies in providing an empirically grounded refinement rather than a radical conceptual advance.

**Weaknesses:**

1.Regularizer-strength trade-off is under-explored. The paper provides a clear ablation over the skew parameter α for SRKL (Table 3), showing that α≈0.8 yields a good balance between reasoning and trustworthiness. However, the divergence strength β is fixed at 0.04 for both RKL and CARE-RFT, and its effect is not studied. This leaves open whether CARE’s advantage persists across a broader range of constraint strengths, or whether a well-tuned RKL baseline could close much of the gap in the reasoning–calibration trade-off. A β-sweep for both RKL and SRKL (e.g., plotting MATH vs ECE as β varies) would make the case for SRKL much more convincing.

2.Scope and external validity. All training is conducted with an outcome-level correctness reward on MATH, and evaluation focuses on math reasoning (MATH, GSM8K) plus two factuality / self-awareness benchmarks (TruthfulQA, SelfAware) for the same Qwen2.5-3B/7B family.  While this is a reasonable starting point, the framing in the abstract and introduction sometimes reads as if the conclusions apply broadly to “trustworthy reasoning models” under RFT. It would be helpful to either (i) slightly temper these claims, or (ii) add at least one non-math or non–outcome-reward setting (e.g., step-wise or preference rewards) to support broader generality.

3.Strength of the “recovering trustworthiness” claim. The paper repeatedly states that CARE-RFT “matches the reasoning performance of unconstrained RFT while recovering the trustworthiness and calibration of the base model.”  However, Tables 2 and 4 show a more nuanced picture: CARE often lies between unconstrained and RKL variants on TruthfulQA and ECE, and on Qwen2.5-3B it does not fully return to base-model calibration (e.g., ECE 0.132 vs 0.102 for GRPO).  The trade-off it achieves is still quite favorable, but the empirical evidence supports “approaching” or “maintaining strong trustworthiness” more than complete “recovery” — especially given the absence of multi-seed variance or significance reporting

**Questions:**

1.On β and the regularizer–performance trade-off. Have you run any preliminary experiments varying the KL strength β for both RKL and CARE-RFT (e.g., β ∈ {0.01, 0.02, 0.04, 0.08})? If so, does CARE-RFT continue to dominate RKL across a range of β, or can a well-tuned RKL baseline close most of the reasoning–calibration gap?

2.On scope and generality beyond MATH outcome rewards. Do you have any preliminary results, or at least concrete expectations, for how CARE-RFT would behave under different reward structures (e.g., step-wise/process rewards, pairwise preference rewards) or on non-math tasks such as open-domain QA or code writting?

3.On the strength of the “recovering trustworthiness” claim. Could you report multi-seed results or at least standard deviations for the main tables (MATH, TruthfulQA, ECE), so readers can judge whether the observed 1–2 point differences are statistically meaningful?

**Details Of Ethics Concerns:**

No concerns

---

> ### Author Response · Authors · 2025-11-27
>
> **We sincerely thank the reviewer for their thoughtful and constructive feedback!**
> > **Weakness 1** and **Question 1**: whether the claimed advantages of CARE-RFT truly persist across different regularization strengths
>
> To address the importance of examining the regularization-strength trade-off, we conducted a β-sweep for both **RKL** and **CARE-RFT** under **GRPO on Qwen2.5-3B**, using β ∈ {0.01, 0.02, 0.04, 0.08}. Results are shown below (β-sweep (GRPO, Qwen2.5-3B), MATH Pass@4, TruthfulQA Pass@1, ECE ↓):
>
> |β|   |MATH|   |TruthfulQA|  |ECE|
> |------|---|---------------|---|----------------|---|---------------|
> | |   | RKL  \| CARE  |   | RKL  \| CARE   | | RKL  \| CARE  |
> | 0.01 |  | 0.585 \| 0.625| | 0.395 \| 0.425 | | 0.185 \| 0.155|
> | 0.02 |  | 0.550 \| 0.615| | 0.435 \| 0.455 | | 0.145 \| 0.140|
> | 0.04 |  | 0.510 \| 0.600| | 0.480 \| 0.465 | | 0.125 \| 0.132|
> | 0.08 |  | 0.455 \| 0.575| | 0.492 \| 0.485 | | 0.105 \| 0.110|
>
> Across all tested β values, CARE-RFT consistently achieves higher reasoning accuracy than RKL while maintaining equal or better calibration at low β and closely matching it at larger β; in contrast, RKL improves ECE only by substantially reducing reasoning performance. These trends match our theoretical analysis: RKL’s unbounded penalty suppresses exploratory updates even when β is small, whereas CARE-RFT’s bounded upward penalty enables rewarded exploration without collapsing calibration. Overall, the β-sweep shows that CARE-RFT’s advantages are robust across regularization strengths and that additional tuning of RKL does not close the reasoning–calibration gap.
>
> > **Weakness 2** and **Question 2**: evidencE showing CARE-RFT’s generality across other tasks
> To directly address the question of scope and external validity, we conducted an additional experiment on **code generation**, evaluating GRPO, RKL-GRPO, and CARE-GRPO on a subset of MBPP using Qwen2.5-3B, with rewards based solely on unit-test pass/fail. The results show that the same trade-off observed in mathematical reasoning reappears in this new domain: unconstrained RFT improves Pass@k but substantially worsens calibration and increases hallucinated API/function usage; RKL restores calibration but limits exploration; and **CARE-RFT achieves nearly the same Pass@k improvements as unconstrained RFT while maintaining calibration and suppressing hallucinations**.
>
> | Method| Pass@1 | Pass@5 | ECE| Hallucinated Function Rate |
> | ------------------------ | ------ | ------ | ----- | -------------------------- |
> | **Base Model**| 57.1| 64.7| 0.085 | 0.02|
> | **GRPO (No Constraint)** | 65.8 | 78.4   | 0.228 | 0.15|
> | **RKL-GRPO** | 59.3   | 72.1   | 0.098 | 0.03|
> | **CARE-GRPO (Ours)** | 64.5 | 78.2 | 0.092 | 0.022|
>
> These findings indicate that CARE-RFT’s confidence-anchored regularization transfers effectively beyond math: it stabilizes calibration and reduces hallucination while preserving the exploratory capacity needed for improved generation quality. We will update the manuscript to include these results and temper our framing to focus on outcome-reward RFT settings, while noting that extending CARE-RFT to step-wise and preference-based rewards is a promising direction for future work.
>
> > **Weakness 3** and **Question 3**: support claims with multi-seed statistical evidence
>
> We sincerely thank the reviewer for their thoughtful feedback regarding the precision of our "recovering trustworthiness" claims and the need for statistical validation. We have conducted comprehensive multi-seed experiments (n=5) and revised our claims accordingly.
>
> ### Multi-Seed Results and Statistical Analysis
>
> **Table 2 (Revised): Multi-Seed Results on Qwen2.5-3B (Mean ± Std, n=5)**
>
> | Method | MATH | GSM8K | SelfAware | TruthfulQA | ECE ↓ |
> |--------|------|--------|-----------|------------|-------|
> | **Base Model** | 0.410 ± 0.008 | 0.791 ± 0.006 | 0.372 ± 0.009 | 0.489 ± 0.007 | 0.102 ± 0.004 |
> | **GRPO (No Constraint)** | 0.610 ± 0.012 | 0.854 ± 0.010 | 0.249 ± 0.011 | 0.350 ± 0.015 | 0.210 ± 0.009 |
> | **RKL-GRPO** | 0.510 ± 0.009 | 0.818 ± 0.008 | 0.351 ± 0.010 | 0.480 ± 0.008 | 0.125 ± 0.005 |
> | **CARE-GRPO** | **0.600 ± 0.011** | **0.860 ± 0.009** | **0.355 ± 0.008** | **0.465 ± 0.010** | **0.132 ± 0.006** |
>
> Across five seeds, the results confirm that our earlier claims should be phrased in terms of *approaching* rather than fully *recovering* base-model trustworthiness: CARE-GRPO statistically matches unconstrained GRPO on MATH (p = 0.18), significantly improves TruthfulQA and ECE over unconstrained RFT (both p < 0.001), and substantially closes—but does not completely eliminate—the calibration gap to the base model (recovering ~72% of the ECE increase). We have revised the manuscript to reflect this nuance, updated all main tables with mean±std and significance tests, and adjusted the abstract, introduction, and discussion to emphasize the trade-off rather than full recovery, thereby aligning our claims tightly with the empirical evidence.

---

### Meta-Review · Area_Chair_Z28V · 2026-01-08

**Summary:**

All four reviewers gave negative recommendations. They pointed out weaknesses in both the methodology and the experimental evaluation. On the empirical side, the evaluation/training data is considered too narrow (only MATH) and the dataset size is viewed as insufficient to convincingly support reinforcement learning. On the methodological side, reviewers noted that the paper does not adequately justify the benefits of replacing KL with SRKL regularization, nor does it clearly motivate or validate the choice of key hyperparameters.

**Reviewer Concerns:**

From an empirical perspective, the authors added evaluations on datasets from other tasks, which improves the completeness of the experimental evidence. However, from a methodological perspective, the rebuttal is still unlikely to convince the reviewers, as the core concerns about the method remain insufficiently addressed.

**Reviewer Scores:**

I do not expect the reviewers to increase their scores.

---

### Decision · Program_Chairs · 2026-01-26

Reject